# Equine Endothelial Cells Show Pro-Angiogenic Behaviours in Response to Fibroblast Growth Factor 2 but Not Vascular Endothelial Growth Factor A

**DOI:** 10.3390/ijms25116017

**Published:** 2024-05-30

**Authors:** Elizabeth J. T. Finding, Ashton Faulkner, Lilly Nash, Caroline P. D. Wheeler-Jones

**Affiliations:** Department of Comparative Biomedical Sciences, Royal Veterinary College, Royal College Street, London NW1 0TU, UK; afaulkner@rvc.ac.uk (A.F.); lnash23@rvc.ac.uk (L.N.); cwheeler@rvc.ac.uk (C.P.D.W.-J.)

**Keywords:** equine, endothelial, angiogenesis, comparative

## Abstract

Understanding the factors which control endothelial cell (EC) function and angiogenesis is crucial for developing the horse as a disease model, but equine ECs remain poorly studied. In this study, we have optimised methods for the isolation and culture of equine aortic endothelial cells (EAoECs) and characterised their angiogenic functions in vitro. Mechanical dissociation, followed by magnetic purification using an anti-VE-cadherin antibody, resulted in EC-enriched cultures suitable for further study. Fibroblast growth factor 2 (FGF2) increased the EAoEC proliferation rate and stimulated scratch wound closure and tube formation by EAoECs on the extracellular matrix. Pharmacological inhibitors of FGF receptor 1 (FGFR1) (SU5402) or mitogen-activated protein kinase (MEK) (PD184352) blocked FGF2-induced extracellular signal-regulated kinase 1/2 (ERK1/2) phosphorylation and functional responses, suggesting that these are dependent on FGFR1/MEK-ERK signalling. In marked contrast, vascular endothelial growth factor-A (VEGF-A) had no effect on EAoEC proliferation, migration, or tubulogenesis and did not promote ERK1/2 phosphorylation, indicating a lack of sensitivity to this classical pro-angiogenic growth factor. Gene expression analysis showed that unlike human ECs, FGFR1 is expressed by EAoECs at a much higher level than both VEGF receptor (VEGFR)1 and VEGFR2. These results suggest a predominant role for FGF2 versus VEGF-A in controlling the angiogenic functions of equine ECs. Collectively, our novel data provide a sound basis for studying angiogenic processes in horses and lay the foundations for comparative studies of EC biology in horses versus humans.

## 1. Introduction

Endothelial cells (ECs) are key players in angiogenesis, the formation of new blood vessels from pre-existing vasculature. This process is important for normal tissue maintenance and repair and is disordered in disease states where endothelial dysfunction is prevalent [1]. Despite growing interest in the horse as a large animal model for a range of diseases, including various musculoskeletal conditions [2,3,4,5], neoplasia [6], cardiac disease [7], retinal disease [8], pulmonary conditions [9,10], endocrinopathies [11], and Raynaud’s disease [12], and the recognition that this species has utility for investigating wound healing [13], regenerative medicine [14,15,16], reproductive [17], immunological [18], and stem cell [16] research, alongside the horse’s extreme athletic ability [19], equine EC biology has received minimal scientific attention. 

During the angiogenic process, ECs are stimulated to migrate, proliferate, and to form the initial structure of the new vessel [20]. Methods by which to examine these processes in vitro are well established using human, bovine, and rodent ECs, with the tube formation assay, scratch wound closure assay, and various proliferation assays providing correlates for differentiation, migration, and proliferation [21,22,23,24]. Vascular endothelial growth factor-A (VEGF-A) is one of the most important growth factors promoting angiogenesis in vivo [25] and is commonly used in in vitro assays as a positive control against which other factors are evaluated. Fibroblast growth factor 2 (FGF2) is also recognised as a potent angiogenic factor [26] and has complex interactions with VEGF-A function [27,28]. However, whether equine ECs respond functionally to key angiogenic factors and in a similar fashion to human ECs remains to be thoroughly investigated. Of the studies employing large vessel equine ECs to date, most have focused on their role in inflammatory conditions [29,30,31], their infection by equine herpes virus [32,33,34], or their involvement in the pathogenesis of laminitis [35,36]. Of the few studies purportedly investigating the angiogenic functions of equine ECs [37,38,39], sufficient characterisation of the isolated cell populations was not performed, and assays recommended by the consensus guidelines for studying angiogenesis in vitro were not always employed [40,41]. Similarly, data from studies using equine blood outgrowth cells (ECFCs) isolated from peripheral venous blood [42,43,44,45,46,47] indicate that these cells have a phenotype that is suggestive of mesenchymal rather than endothelial lineage [48], making these cells unsuitable as surrogates for equine vascular ECs and for studying angiogenic processes. 

In this study, we have, for the first time, developed and optimised methods for the isolation, culture, and purification of equine aortic ECs (EAoECs). To allow for thorough investigation of their angiogenic functions, we have optimised standard in vitro angiogenesis assays, which we employ routinely in human ECs [21,22,23,24,40] for use with equine ECs, and have investigated their functional responses to a range of angiogenic growth factors. 

## 2. Results

### 2.1. ECs Can Be Isolated from Equine Aorta by Mechanical Dissociation

Understanding of the vascular endothelial function and dysfunction in horses is hampered by the current lack of robust, well characterised in vitro models for mechanistic research. In our initial studies, we sought to optimise both the isolation method and culture conditions for EAoECs. Intimal cells were isolated from fresh equine aortas via either mechanical scraping or collagenase digestion and immunofluorescent staining for von Willebrand factor (VWF) expression used to classify cultures as EC ‘rich’ (>65% VWF positive cells) or EC ‘poor’ (<65% VWF positive cells) (Figure 1). Figure 1a shows the number of cell isolates classified as EC rich or poor derived from mechanical scraping versus collagenase treatment. Mechanical scraping was the superior isolation method, resulting in a higher proportion of isolates with >65% VWF-positive cells. We then optimised the culture conditions for EC growth by comparing the rates of proliferation in a standard growth medium used routinely for culturing primary human ECs (M199 supplemented with 20% equine serum and endothelial cell growth factor; M199 [49]) and in a medium designed to support expansion of commercially available ECs (EGM2, supplemented with 20% equine serum; Appendix A). EAoECs cultured in EGM2 proliferated at a higher rate than cells cultured in M199, as assessed by nuclear staining (Figure 1b). However, the presence of the additives in complete EGM2 resulted in cells with a more elongated morphology than those cultured in EGM2 without additives (EBM2), which displayed the classical cobblestone morphology expected for confluent ECs in static culture (Figure 1c). Cells in both EGM2 and EBM2 expressed the EC marker VWF. (Figure 1c). These results indicate that EBM2 is the optimal medium for culture of EAoECs for investigation of angiogenesis since these cells exhibit a more quiescent phenotype. The effect of serum concentration on growth rate was examined under these conditions. Equine serum at 20% resulted in the highest growth rate (Appendix A). All subsequent experiments were performed with cells acquired via mechanical scraping (Figure 1d) and cultured in EBM2 with 20% equine serum.

Although cell isolation via mechanical scraping was superior to collagenase digestion, significant contamination of isolated populations by non-ECs (cells lacking VWF) remained. To address this, we applied a magnetic cell sorting method to enrich the isolates for ECs. Since there are no commercially available antibodies recognising equine EC surface antigens, antibodies targeting ECs in other species were investigated for cross reactivity. Multiple antibodies against known EC markers (CD31, VE-cadherin, VEGFR2, Claudin-5, ZO-1, Ang-2) were not cross reactive (Appendix A). However, one anti-VE-cadherin antibody (clone 55-7H1; ThermoFisher) was confirmed to recognise equine VE-cadherin on the cell membrane in immunocytochemical analysis of cultured EAoECs (Figure 2a), in en face staining of equine intercostal artery sections (Figure 2b), and via flow cytometry of EAoECs stained in suspension (Figure 2c). The flow cytometry analysis showed that 99% of the stained population was positive for VE-cadherin. Immunofluorescence of the same isolate of fixed cells using an anti-VWF antibody showed that 98% of the population was positive for VWF, indicating excellent correlation between the two markers.

Having identified a reliable equine EC-targeted antibody, we then used this to sort EAoECs from mixed vascular cell isolations from equine aortas (Figure 2d). As expected, a high proportion of ECs (VWF-positive cells >95%) was present in sorted cell isolates compared to unsorted isolates (Figure 2e). Further studies using an antibody against the smooth muscle cell marker, smoothelin, showed that vascular smooth muscle cells are likely the predominant contaminating cell type in mixed cell populations isolated via mechanical scraping (Appendix A).

### 2.2. EAoECs Respond to FGF2 but Not VEGF-A Stimulation

We next performed experiments using sorted cell populations enriched for ECs to investigate the functional responses of EAoECs to a range of growth factors known to exert pro-angiogenic effects on human and rodent ECs. The angiogenic potential of epidermal growth factor (EGF), FGF2 (referred to throughout as FGF), insulin-like growth factor (IGF), and VEGF-A was explored by measuring their individual effects on tubulogenesis and the rate of EAoEC proliferation. Growth factors were used at concentrations shown to be maximally effective in similar studies in human ECs. Initial experiments used recombinant forms of the human growth factors, except for VEGF-A, where recombinant equine VEGF-A was employed. As shown in Figure 3a, FGF significantly enhanced tubulogenesis by EAoECs; EGF, IGF, and VEGF-A caused no significant increase in comparison to control. In experiments performed using cells cultured in EBM2 with endothelial cell growth supplement, there was no significant tube formation response above baseline to pro-angiogenic growth factors, confirming the importance of using EBM2 for culture and functional investigation of these cells (Appendix A). EGF, FGF, and IGF all increased the rate of EC proliferation, with FGF treatment promoting the greatest effect (Figure 3b). Since pro-angiogenic growth factors act, at least in part, through mitogen activated protein kinase - extracellular signal-regulated kinase (MEK-ERK) signalling pathways, ERK1/2 phosphorylation status was examined via Western blotting. FGF was the most effective stimulant of ERK1/2 phosphorylation (Figure 3c). Unexpectedly, no effect of VEGF-A treatment on proliferation, tube formation, or ERK1/2 activity was observed. Collectively, these results indicate that of the growth factors tested, EAoECs are most sensitive to FGF stimulation.

### 2.3. The Pro-Angiogenic Effect of FGF2 Is Dependent on FGFR1/MEK-ERK Signalling

To further explore the pro-angiogenic actions of FGF on equine ECs, we next determined the maximally effective concentration using recombinant equine FGF. These studies used FGF concentrations ranging from 1 to 100 ng/mL, depending upon the functional readout (Figure 4). Significant enhancement of tubulogenesis above basal was evident at the lowest FGF concentration examined (5 ng/mL) with the maximal response observed at 20 ng/mL (Figure 4a). Increased proliferation was seen at very low concentrations of FGF, with significant increases in cell number observed between 1 and 20 ng/mL, and a maximal response observed at 5 ng/mL FGF (Figure 4b). We additionally used the scratch wound closure assay to assess the effect of FGF on directional EAoEC migration. We saw significant increases in proportional wound closure in the presence of FGF at concentrations between 10 and 100 ng/mL, with 92% closure at the highest concentration (Figure 4c). To determine whether wound closure resulted from EC migration or proliferation, we measured proliferation in response to FGF over a similar time period; there was no significant FGF-induced proliferation after 18 h (Appendix A). To evaluate the effect of equine FGF on ERK1/2 phosphorylation, we first determined the appropriate stimulation time using FGF at 20 ng/mL, which had a robust stimulatory effect in all the functional assays. As shown in Figure 4d (upper panel), the maximal response to FGF was seen after a 10-min stimulation. Further experiments then measured ERK1/2 phosphorylation at this optimised time point and revealed significantly increased phosphorylation between 5 and 40 ng/mL FGF, with the maximal response at 5 ng/mL (Figure 4d, lower panels).

Having established a dominant functional role for FGF in equine ECs, we next investigated the involvement of FGF receptor 1 (FGFR1) in these responses using the pharmacological FGFR1 inhibitor, SU5402 [50]. Treatment with SU5402 (10 µM) completely inhibited FGF-induced tubulogenesis (Figure 5a) and scratch wound closure (Figure 5b), partially reduced FGF-stimulated proliferation (Figure 5c), and prevented the increase in ERK1/2 phosphorylation in FGF-challenged cells (Figure 5d). Inhibition of the MEK-ERK pathway using the MEK1/2 inhibitor PD184352 (10 µM) blocked FGF-driven tube formation (Figure 5a) and reduced wound closure (Figure 5b), proliferation (Figure 5c), and ERK1/2 phosphorylation (Figure 5d) in comparison to control in the absence of FGF. Collectively, these data suggest that FGF’s pro-angiogenic effects on EAoECs are mediated by FGFR1 and downstream MEK-ERK signalling.

### 2.4. Lack of Sensitivity of EAoECs to VEGF-A May Be Due to Differences in Receptor Expression

VEGF-A exerts well-documented pro-angiogenic actions on ECs from human and other commonly studied species [22,23,49], so the inability of EAoECs to respond effectively to VEGF-A in vitro at a standard concentration (25 ng/mL; Figure 3) was an unexpected finding. ERK1/2 phosphorylation in human ECs (HUVEC) was increased to a similar extent by both equine and human VEGF-A, confirming that equine recombinant VEGF-A is biologically active (Appendix A). To confirm that EAoECs do not respond to VEGF-A, the effects of a range of VEGF-A concentrations on functional responses and the MEK-ERK pathway activation were tested. VEGF-A (1–50 ng/mL) had no stimulatory effect on tube formation (Figure 6a), scratch wound closure (Figure 6b), rate of proliferation (Figure 6c), or ERK1/2 phosphorylation (Figure 6d) at any of the concentrations examined. 

To determine whether this lack of response could reflect the absence or poor expression of VEGF receptors by EAoECs, we measured growth factor receptor gene expression in EAoECs and compared this to receptor expression in HAoECs and HUVECs using qPCR (Figure 7). These studies showed that levels of VEGFR1 and 2 were much lower than FGFR1 in EAoECs (Figure 7a). In contrast, VEGFR1 and 2 were expressed at similar or greater levels than FGFR1 in both human EC types (Figure 7b,c). EAoECs, in common with HAoECs and HUVECs, strongly expressed the VEGF co-receptor, neuropilin 1 (NRP1) (Figure 7a). These findings, in combination with the lack of responses to VEGF-A, indicate that there are differences in the pro-angiogenic roles of VEGF-A and FGF in EAoECs in comparison to human ECs.

## 3. Discussion

Assessment of EC angiogenic functions in vitro is crucial for understanding the molecular control of angiogenesis and enabling translation to regenerative and disease settings. In this study, we have developed methods for assessing the angiogenic behaviours of equine aortic ECs and have identified differences in the responses to pro-angiogenic growth factors between equine and human ECs. 

We optimised methods for the isolation, enrichment, and culture of EAoECs and found that mechanical isolation, followed by positive selection via magnetic cell sorting, was the most effective approach for obtaining EC-rich isolates from equine aortas. Importantly, as part of the development of this method, we identified a specific monoclonal anti-VE-cadherin antibody that can detect the equine antigen on the EC surface in cultured cells and in equine vessels en face. To our knowledge, this is the only report of a commercially available antibody suitable for this purpose in equine cells. Other methods for obtaining equine ECs have been reported previously but the purity of these populations was either not evaluated or contamination with vascular smooth muscle cells was noted [30,31,38,51,52]. CD31 is used in tissue sections to visualise equine ECs using immunohistochemistry [53,54], but none of the antibodies tested in this study were cross-reactive with EAoECs in culture using immunocytochemistry, presumably due to differences in antigen presentation (see Appendix A). An attempt to purify equine ECs from mixed cultures has been described using an anti-CD31 antibody, but the success, or otherwise, of this approach was not assessed, and the cell images provided in the manuscript show persistent contamination with non-ECs [55]. An alternative, non-immunological method, based on the uptake of fluorescently conjugated acetylated low-density lipoprotein, has also been used in an attempt to purify equine ECs [42,56]. However, this marker cannot be considered specific for ECs since it can also be taken up by mesenchymal stromal cells and is therefore unsuitable for defining purity [57].

Following optimisation of EAoEC isolation and culture, we investigated their pro-angiogenic functions by assessing proliferation, migration, and tubulogenesis following growth factor stimulation and the involvement of MEK-ERK signaling in these responses [26]. We showed that a MEK1/2 inhibitor, PD184352, blocked FGF-induced ERK1/2 phosphorylation, confirming that FGF treatment enhances ERK1/2 phosphorylation in a MEK-dependent manner. Inhibiting MEK activity also suppressed FGF-driven tube formation, scratch wound closure, and proliferation, suggesting that MEK-ERK pathway activity is a key regulator of the functional changes induced by FGF in equine ECs. An FGFR1 inhibitor reduced proliferation, migration, and tube formation in FGF-stimulated ECs, indicating that the pro-angiogenic effects of FGF are dependent on FGFR1 activity. Together, these data show that FGFR1 and downstream coupling to MEK-ERK signalling drive angiogenic changes in equine ECs, indicating a key role for FGF in equine EC function.

Unexpectedly, we found that VEGF-A, a well-established pro-angiogenic growth factor in vitro and in vivo, had no stimulatory effect on the angiogenic functions of equine ECs in vitro. The reasons for this lack of sensitivity are unknown and likely multifactorial, but one potential explanation would be low or absent VEGF receptor/co-receptor expression. We measured mRNA expression for growth factor receptors and found that FGFR1 is expressed at a relatively higher level than both VEGFR1 and 2 in EAoECs, in contrast to HAoECs or HUVECs, which both express similar levels of the different receptors. NRP1, a key co-receptor for VEGFR2, was strongly expressed in both equine and human ECs. It is feasible that the lower relative expression of VEGFR1 and VEGFR2 could account, at least in part, for the lack of response to VEGF-A seen in EAoECs in functional assays of angiogenic potential and in the assessment of MEK-ERK signaling. The angiogenic responses of equine ECs to FGF and VEGF-A have not been studied previously, so there is no body of work in this area. In addition, the limited investigations of equine EC angiogenic potential to date have not used the accepted methods for evaluating these behaviours [38], or they used cells which had not been properly characterised [37,38,39,42,55]. Treatment of equine limb wounds with a combination of IL-10 and VEGF-E led to an increase in the number of blood vessels (assessed using anti-CD31 and collagen IV staining) within the granulation tissue, although it is notable that the effect of the two factors individually was not assessed, so the mediator of the apparent pro-repair effect is not clear [58]. There is little understanding, even in humans, of the significance of VEGF-E signalling and its EC-directed effects for tissue repair/angiogenesis. To date, FGF has not been investigated for its pro-angiogenic effects in any in vivo setting in the horse. Further in vivo and in vitro work, in both macro- and micro-vascular settings, is now required to explore the role of VEGF-A and VEGF receptors in equine endothelial cells and their pathophysiological significance. Responses to growth factors may also be influenced by stimulatory or inhibitory factors within the equine serum. The functional assays reported here were all performed in a low serum medium (1–5%). However, whilst short-term signalling measurements (e.g., ERK phosphorylation) can be performed under serum-free conditions, functional assays (tubulogenesis, scratch wound) cannot because the absence of serum severely impacts cell viability.

Both FGF and VEGF-A are potent pro-angiogenic stimuli for human ECs [21,22,59,60,61], and receptors for these growth factors are expressed at similar levels in human ECs [62]. In the horse, levels of FGFR1, FGFR2, and VEGFR2 expression have been studied in the oviduct at different sites and at different stages of the estrous cycle [63]. At each site and each time point, the mRNA expression level, relative to β2-microglobulin, was lower for VEGFR2 than for either FGFR1 or FGFR2, which is in agreement with the lower VEGFR2 expression seen in EAoECs in this study. The lack of growth factor receptor antibodies cross-reactive with the equine proteins precludes detailed investigation of FGFR1 and VEGF receptor function in equine ECs. However, the receptor expression pattern, the predominant effects of FGF, and the VEGF-A insensitivity revealed in this study raise the possibility that regulation of angiogenesis in the horse differs from that in humans. The cross-talk between FGF and VEGF signaling is highly complex [64], and there is evidence from studies in bovine ECs in vitro and mouse models in vivo that FGF signaling is essential for VEGF function [27] and that VEGF-A and FGF2 synergistically stimulate angiogenesis in vitro [28,65] and in vivo [66]. Whether similar interactions regulate the angiogenic functions of equine ECs and the relevance of these for angiogenesis in vivo remain to be determined.

This work advances the field of equine EC research and provides a strong foundation for scientifically sound and meaningful comparative investigations of EC function and angiogenesis in horses and in humans. 

## 4. Materials and Methods

### 4.1. Endothelial Cell Isolation and Culture

Equine aortic endothelial cells (EAoECs) were isolated from mixed-breed adult male and female horses (n = 143 over entire duration of study) euthanised at a commercial abattoir for reasons other than research. Aortas were cut distal to the aortic arch and removed from the thoracic cavity by transecting the intercostal arteries and the aorta at the level of the diaphragm. Vessels were immediately placed in individual sterile containers, immersed in culture medium (Medium-199 with Hank’s balanced salts, M199H; supplemented with 100 U/mL penicillin and 100 U/mL streptomycin), and kept on ice for transport to the laboratory. In a class II safety cabinet, aortas were cleaned of connective and adipose tissue via blunt dissection and incised longitudinally between the paired intercostal artery openings. The luminal surface was examined for lesions (e.g., calcification indicative of parasite migration) and discarded if any were present. 

### 4.2. Cell Isolation by Scraping

The luminal surface was gently scraped with the back of a sterile scalpel blade, avoiding the peripheral sections of the aorta. The accumulated material on the scalpel blade was transferred to a sterile 15 mL centrifuge tube and incubated (37 °C, 5% CO_2_) with 3 mL collagenase solution to dissociate the material into individual cells (0.25 mg/mL in endothelial cell basal medium 2; EGM2; Promocell, Heidelberg, Germany; sterile filtered) for 10–20 min. The collagenase solution was then diluted with an equal volume of EGM2 prior to centrifugation at 300× *g* for 5 min at 20 °C. The supernatant was discarded, and the cell pellet was resuspended in 6 mL of EGM2 supplemented with 20% horse serum, 100 U/mL penicillin, and 100 U/mL streptomycin (equine basal medium, (EBM)). The cell suspension was transferred to a gelatin-coated (1% (*v*/*v*)) tissue culture flask (25 cm^2^), and cells were maintained in a humidified tissue culture incubator (37 °C, 5% CO_2_). The culture medium was replaced in full every 2–3 days. Once confluent (2–5 days), cell cultures were purified using magnetic-activated cell sorting (MACS). Following sorting, cells were grown on gelatin-coated flasks (75 cm^2^), and the culture medium (12 mL/flask) was replaced every 2–3 days. Once confluent, cells were plated onto the appropriate tissue culture plates/slides for experimentation and were used between passage 1 and 5.

### 4.3. Optimisation of Isolation and Culture Conditions for EAoECs

A detailed description of equine cell isolation using collagenase and methods for optimising culture conditions are provided in the Appendix A.

### 4.4. Magnetic-Activated Cell Sorting

Magnetic-activated cell sorting (MACS) was performed using the CELLection Pan Mouse IgG Kit (ThermoFisher, Waltham, MA, USA) following the manufacturer’s protocol (direct technique). Magnetic beads were conjugated following the manufacturer’s guidelines using 1 μg VE-cadherin monoclonal antibody (clone 55-7H1, ThermoFisher) per 100 μL beads. Confluent EAoECs were detached using trypsin-EDTA (0.05%), centrifuged (300× *g*, 5 min, 4 °C), and resuspended in 1 mL buffer (Ca^2+^- and Mg^2+^-free phosphate-buffered saline (PBS) with 0.1% bovine serum albumin (BSA) and 2 mM EDTA). Conjugated beads (25 μL) were added to the cells in a 15 mL centrifuge tube. The tube was incubated at 4 °C on a roller for 20 min then placed in a magnet (DynaMag-15, ThermoFisher) for 2 min to allow the magnetic beads and bound cells to migrate to the sides of the tube. The remaining cell suspension was aspirated and either discarded or transferred to a tissue culture flask for continued culture. The tube was removed from the magnet, cells were resuspended in buffer (Ca^2+^- and Mg^2+^-free PBS with 0.1% BSA, pH 7.4), and the tube was returned to the magnet. This washing process was repeated twice. The cells were resuspended in EGM2 supplemented with 1% horse serum, 1mM CaCl_2_, 5 mM MgCl_2_, and 2% DNase (pH 7.0–7.4, 37 °C, 200 μL) and then incubated on a roller at 20 °C for 15 min to detach the beads from the sorted cells (beads were not removed from the suspension following incubation since this process led to loss of cells in preliminary experiments). Following incubation, the cell suspension was diluted in EBM (final volume 12 mL) and transferred to gelatin-coated flasks for continued culture.

### 4.5. Human Endothelial Cell Isolation and Culture

Human umbilical cord collection (obtained with informed written consent) and the use of human endothelial cells conformed to the principles outlined in the Declaration of Helsinki and were approved by the NHS Health Research Authority East of England—Cambridge South Research Ethics Committee (REC reference 16/EE/0396). All experiments were performed in accordance with relevant guidelines and regulations. Human umbilical vein endothelial cells (HUVECs) were isolated and cultured as described previously and were used between passage 2 and 4 [21]. Cells were cultured on 1% gelatin and maintained in M199E growth medium (with M199E containing endothelial cell growth factor (20 μg/mL) and 20% (*v*/*v*) FBS). 

Human aortic endothelial cells (HAoECs) were maintained in EGM-2 according to the supplier’s instructions (Lonza, Visp, Switzerland) and used for experiments at passages 4–8.

### 4.6. Assessment of Cell Morphology

Cells were visualised using phase contrast imaging to assess cell morphology using a DMIRB inverted microscope (Leica Microsystems, Milton Keynes, UK) and an MRm monochrome camera controlled through Zen software (V2.6; Carl Zeiss Ltd., Cambridge, UK). Cell morphology was quantified by manually measuring the cell length (longest diameter measurement) and width (shortest diameter measurement) using Fiji (ImageJ v 2.1.0) software. Measurements were repeated for 10 representative cells for each condition. 

### 4.7. Endothelial Cell ‘Tube’ Formation Assay

The dynamic behaviour of EAoECs on the extracellular matrix, reflective of their angiogenic potential, was investigated with a modified EC tube-forming assay using a thin layer of matrix [21]. Sub-confluent (70–95%) EAoECs in a T75 flask were serum-deprived in EGM2 + 1% horse serum for 1 h, trypsinised, and re-suspended at 120,000 cells/mL. The wells of a 96-well plate were coated with 2 µL/well of Geltrex™ (Life Technologies, Carlsbad, CA, USA) using the insert of a sterile Eppendorf Combitip^®^ and left to set at 37 °C for 30 min. EAoECs were plated onto the coated wells (50 µL/well; 6000 cells/well) before addition of an equal volume of 2 × concentrated experimental treatment (in sextuplet, made up in EGM2 + 1% horse serum). EAoECs were incubated overnight for 16 h, and the centre of each well was imaged using a DMIRB inverted microscope (×10 magnification; as above). The number of branches was quantified manually using Fiji (ImageJ v 2.1.0) software and displayed graphically as ‘tube count’.

### 4.8. Scratch Wound Assay

EAoECs were plated onto gelatin-coated 48-well plates (40,000 cells/well) and grown to confluence. A single vertical scratch was made in the centre of each well using a sterile 200 µL pipette tip. Cells were then washed in warm PBS before the addition of treatments (in triplicate; diluted in EGM2 + 5% horse serum). The midpoint of each scratch was imaged using a DMIRB inverted microscope (×5 magnification, as above). Cultures were incubated overnight for 18 h and imaged again. Percentage wound closure following each treatment was calculated by manually measuring the area of the wound in the field of view at time 0 and at 18 h using Fiji (ImageJ v 2.1.0) software. 

### 4.9. Measurement of Endothelial Cell Proliferation 

EAoECs were plated onto gelatin-coated 96-well plates (5000 cells/well) and left to adhere in EBM. After 4 h, the growth medium was aspirated and replaced with experimental treatments (in triplicate, made up in EGM2 + 5% horse serum), which were refreshed every 24 h. After 24, 48, and 72 h, the treatments were removed. The cells were washed in PBS and fixed in 4% paraformaldehyde (PFA) for 15 min. After further washing, nuclei were stained with Hoechst 33342 (0.5 μg/mL in PBS) for 10 min. The centre of each well was imaged using a DMIRB inverted microscope (×10 objective; as above). The number of nuclei was quantified using an automated cell counter (Fiji (ImageJ v 2.1.0) software), and the increase in cell number between 4 and 72 h was calculated.

### 4.10. Western Blotting and Immunofluorescence

Western blotting on whole EAoEC lysates was performed as described previously for human ECs, with some modifications [49]. Details are provided in the Appendix A. 

Protein expression was assessed via immunofluorescence in cells cultured on gelatin-coated 96-well plates or cells cultured on collagen-coated 9 mm coverslips. Cells were fixed in 4% PFA (15 min), washed (to remove unbound PFA; 50 mM ammonium chloride, 15 min), permeabilised (0.1% Triton-X in PBS, 5 min), and blocked with PGAS (phosphate gelatin and saponin solution; 0.2% gelatin, 0.02% saponin, and 0.02% sodium azide in PBS; 5 min) prior to incubation with the primary antibody (1 h; Appendix A), followed by the fluorescent-conjugated secondary antibody and nuclear stain (Hoechst 33342; 0.5 µg/mL). Coverslips were mounted using Mowiol (Sigma Aldrich, Dorset, UK). 

Cells in 96-well plates were imaged using a DMIRB inverted microscope (×20 objective; as above). Cells on coverslips were imaged using an SP8 confocal microscope (Leica Microsystems, Milton Keynes, UK) controlled through LAS-X software (V3.5; Leica Microsystems, Milton Keynes, UK).

### 4.11. En Face Imaging of Equine Intercostal Artery

Short (5–10 mm) segments of intercostal artery were transected from the thoracic aorta following EC isolation from the aortic tissue. Intact vessel segments were placed in individual wells of a 96-well plate for staining, orientated vertically to ensure the lumen was open, and volumes of all solutions were adjusted to ensure complete coverage of the segment (approximately 200 µL). Segments were washed (PBS), fixed (4% PFA, 12–24 h, 4 °C), washed (50 mM ammonium chloride, 1 h), permeabilised (0.1% Triton-X in PBS, 1 h), and blocked against non-specific binding (3% BSA, 12–24 h, 4 °C) prior to incubation with the primary antibody overnight (4 °C; Appendix A), followed by the fluorescent-conjugated secondary antibody. Artery segments were cut longitudinally to obtain flat square sections of tissue (approximately 5 mm × 5 mm) and mounted luminal-side-down in a glass-bottomed 30 mm dish with compression applied from the serosal side. PBS was added to the dish to maintain hydration. Sections were imaged using an SP8 confocal microscope (as above).

### 4.12. Flow Cytometry

Cells were trypsinised, washed, and resuspended in 1% BSA in PBS at 1 × 10^6^ cells/mL. Aliquots of cells (50 µL) were incubated with fluorescent-conjugated antibody or appropriate controls (see Appendix A) for 30 min on ice, protected from light. Cells were washed twice (1% BSA, 4 °C, 5 min, 400× *g*) and resuspended to a final volume of 500 µL before immediate analysis. Flow cytometry was performed using a Canto II flow cytometer (BD) with Diva software version 8.0.1. The instrument was calibrated with cell tracker beads (BD). Cells were located in a side-scatter (logarithmic scale, *y*-axis) and forward-scatter (logarithmic scale, *x*-axis) plot. Gated cells were displayed in a forward-scatter height (linear scale; *y*-axis) and forward-scatter area (linear scale, *x*-axis) plot to exclude doublets. Single-cell events were then displayed on a histogram of fluorescence intensity, with the isotype control sample distribution overlying the antibody-stained population to identify the stained population.

### 4.13. qPCR

RNA was extracted using the Qiagen RNeasy Plus Mini-Kit according to the manufacturer’s instructions. Complementary DNA (cDNA) was synthesised using a High-Capacity cDNA Reverse Transcription kit (ThermoFisher, Oxford, UK). Endothelial gene expression was measured via RT-qPCR (Appendix A) using SYBR^®^ Green JumpStart™ Taq ReadyMix™ (Sigma Aldrich; Appendix A). Primers were designed using Primer3 (version 4.1.0) and NCBI Primer-BLAST software (https://www.ncbi.nlm.nih.gov/tools/primer-blast/index.cgi?GROUP_TARGET=on, accessed 7 September 2022; Appendix A). PCR reactions were analysed using CFX Manager 3.1 (Bio-Rad, Hercules, CA, USA), and analysis was performed using the 2^−ΔCt^ method with results normalised to the β-actin internal control gene.

### 4.14. Statistical Analysis

Statistical/graphical analysis was performed using GraphPad Prism version 9 (GraphPad Software). Unless otherwise stated, data are presented as mean ± standard error of the mean (SEM) from independent experiments performed on cell isolates from separate horses. A *p*-value of <0.05 was considered significant. Data were analysed as specified in the figure legends via *t*-test, ANOVA, or a mixed-effects model, as appropriate. Normality was assessed using the Shapiro–Wilk test for normal distribution.

## Figures and Tables

**Figure 1 ijms-25-06017-f001:**
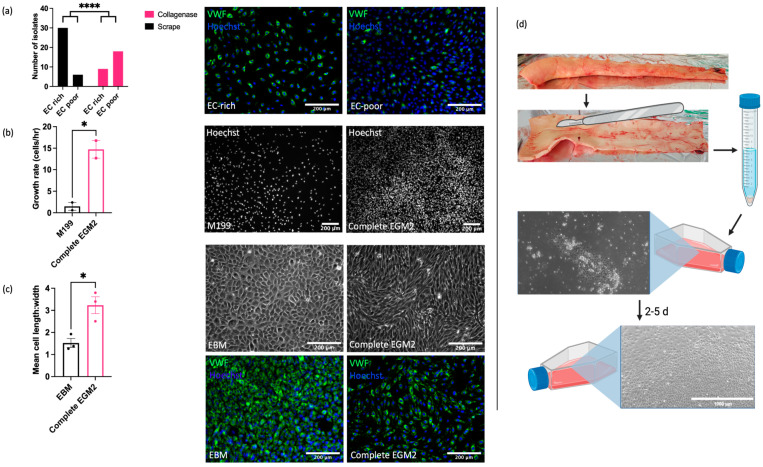
Optimisation of EAoEC isolation and culture. (**a**) Left: mechanical scraping yielded a higher proportion of EC-rich cultures (defined as >65% VWF-positive cells) than collagenase digestion (**** *p* < 0.0001; Fisher’s exact test). Data are from a total of 63 isolations. Right: representative images of EAoEC isolates fixed and processed for VWF immunofluorescence (green). (**b**) Left: EAoECs proliferated at a faster rate when cultured in complete EGM2 than in M199 (* *p* = 0.03; *t*-test; mean ± S.E.M for *n* = two biological replicates with three technical repeats). Right: EAoECs were fixed, and nuclei were stained (Hoechst) for cell counting after 72 h. (**c**) Left: EAoECs were cultured in complete EGM2 or EGM2 without additives (EBM), and cell morphology was evaluated via light microscopy (* *p* = 0.02; *t*-test; mean ± S.E.M, n = three isolates). Right upper: representative phase contrast images of EAoECs cultured to confluence in complete EGM2 or EBM. Right lower: representative images of EAoEC cultured in complete EGM2 or EBM, fixed and processed for VWF immunofluorescence (green). (**d**) Final EAoEC isolation strategy: equine aortas were cleaned of connective and adipose tissue via blunt dissection and incised longitudinally between the paired intercostal artery openings. The luminal surface was scraped with the back of a sterile scalpel blade. The accumulated material on the scalpel blade was transferred to a sterile 15 mL centrifuge tube and incubated with collagenase solution to dissociate the material into individual cells. The cell pellet was transferred to a gelatin-coated tissue culture flask (×10 magnification) and cultured to confluence. Figure created with BioRender.com (https://app.biorender.com/user/signin, accessed 14 September 2023).

**Figure 2 ijms-25-06017-f002:**
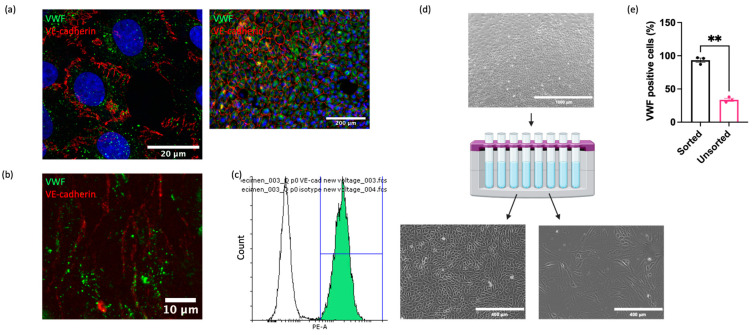
Purification of EAoEC populations. EAoECs were purified using magnetic cell sorting. (**a**) High-power confocal (×63; left) and low-power widefield (×20; right) images of EAoECs immunostained for cytoplasmic VWF (green) and membranous VE-cadherin (red). (**b**) Maximum projection of a z-stack confocal image of equine intercostal artery en face showing immunodetection of cytoplasmic VWF (green) and membranous VE-cadherin (red). (**c**) Live EAoECs were incubated with a PE-conjugated anti-VE-cadherin antibody and analysed via flow cytometry for surface VE-cadherin expression. VE-cadherin-stained cells (green frequency distribution) exhibited a brighter fluorescent signal than cells incubated with the isotype control antibody (white frequency distribution). (**d**) Unsorted cells were subjected to magnetic cell sorting using magnetic beads conjugated to an anti-VE-cadherin antibody, resulting in the positive population showing the classic cobblestone endothelial morphology and the negative population showing morphology consistent with vascular smooth muscle cells (created with BioRender.com (https://app.biorender.com/user/signin, accessed 14 September 2023)). (**e**) The proportion of VWF-positive cells was greater in the sorted population than the unsorted population, as assessed by immunofluorescence (** *p* = 0.008; *t*-test; mean ± S.E.M, *n* = 3).

**Figure 3 ijms-25-06017-f003:**
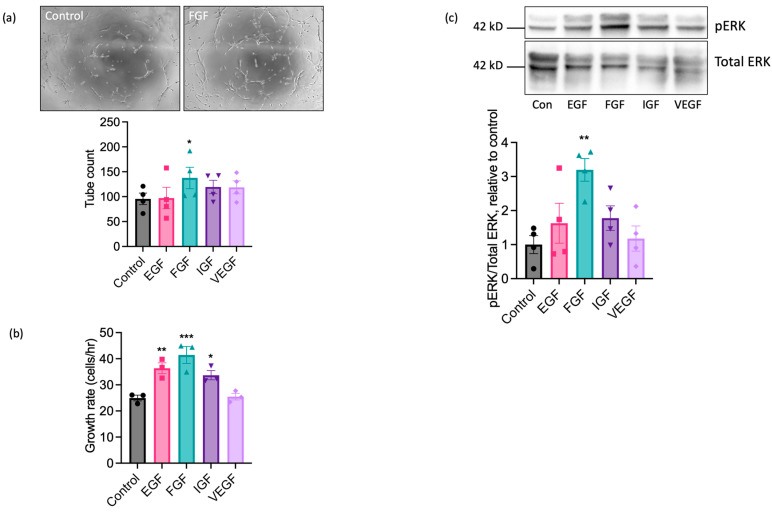
FGF2 is a potent pro-angiogenic factor for EAoECs. (**a**) EAoECs were exposed to FGF2 (10 ng/mL), epithelial growth factor (EGF; 10 ng/mL), insulin-like growth factor (IGF; 50 ng/mL), or VEGF-A (25 ng/mL) for 16 h, and tube formation was analysed by manually counting the number of branches (* *p* = 0.02; repeated measures one-way ANOVA with Dunnett’s multiple comparisons test compared to control; mean ± S.E.M, *n* = 4). Images are representative of tubulogenesis induced by FGF-2 at 16 h (×10 magnification). (**b**) Effects of growth factors on EAoEC proliferation rate over 72 h (* *p* = 0.04, ** *p* = 0.009, *** *p* = 0.0007; one-way ANOVA with Dunnett’s multiple comparisons test compared to control; mean ± S.E.M for *n* = three technical replicates). (**c**) EAoECs were exposed to growth factors for 10 min and ERK phosphorylation assessed by immunoblotting using a phospho-specific ERK1/2 antibody. Blots were stripped and re-probed for total ERK1/2, and all blots analysed by densitometry. Data are mean ± S.E.M (*n* = four independent experiments on separate EAoEC isolates). ** *p* = 0.007; repeated measures one-way ANOVA with Dunnett’s multiple comparisons test compared to control. Representative cropped immunoblots are shown. Concentrations of growth factors used were consistent across experiments.

**Figure 4 ijms-25-06017-f004:**
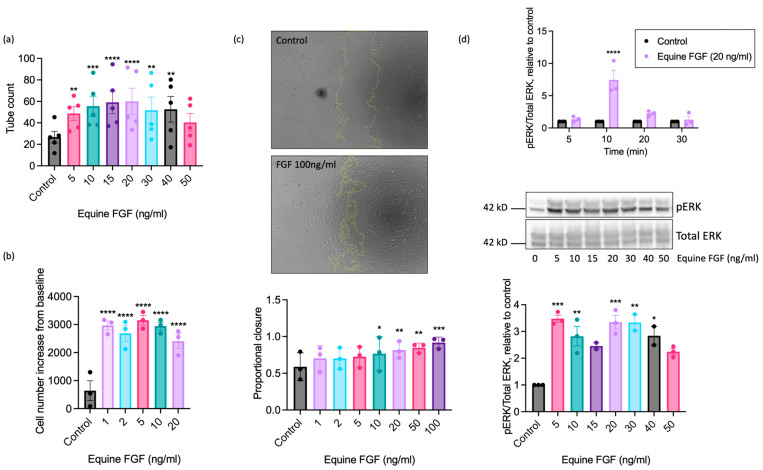
Concentration-response characteristics of EAoECs to equine recombinant FGF2. Tubulogenesis (panel (**a**); *n* = 5), proliferation (panel (**b**); *n* = 3), scratch wound closure (panel (**c**); *n* = 3), and ERK1/2 phosphorylation at 10 min (panel (**d**); *n* = 3) were evaluated in response to the indicated concentrations of FGF2. The concentration ranges used were designed to identify the maximal response. In the scratch wound closure experiments, the FGF2 concentration was increased to 100 ng/mL after preliminary experiments, indicating that concentrations below 50 ng/mL did not lead to a maximal response. Representative cropped immunoblots are shown. The images in panel c are representative and show control and FGF2-stimulated wounds after 18 h (×5 magnification). Data are mean ± S.E.M and were analysed via repeated measures one-way ANOVA with Dunnett’s multiple comparisons test (**** *p* < 0.0001, *** *p* < 0.001, ** *p* < 0.01, * *p* < 0.05 versus unstimulated control).

**Figure 5 ijms-25-06017-f005:**
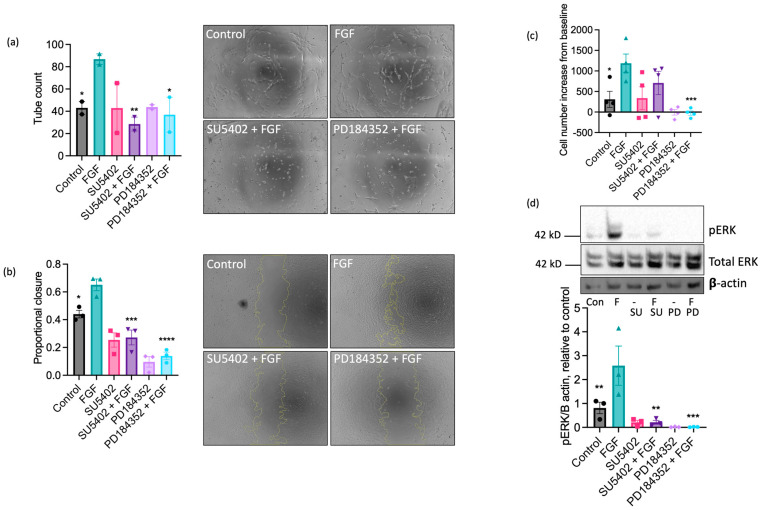
The pro-angiogenic effects of FGF2 are mediated by FGFR1 and MEK-ERK signalling. Tube formation ((**a**); *n* = two biological replicates with six technical repeats; FGF2, 20 ng/mL), scratch wound closure ((**b**); *n* = three biological replicates; FGF2, 100 ng/mL), proliferation ((**c**); *n* = four biological replicates; FGF2, 5 ng/mL), and ERK/2 phosphorylation ((**d**); *n* = three biological replicates; FGF2, 5 ng/mL) were measured in control and FGF2-stimulated (at a concentration to give maximal response (see Figure 4)) EAoECs in the absence or presence of SU5402 (FGFR1 inhibitor, 10 µM) or PD184352 (MEK1/2 inhibitor, 10 µM). Representative images are shown for tubulogenesis (×10 magnification) (**a**) and scratch wound closure (×5 magnification) (**b**), as well as representative immunoblots for pERK, total ERK expression, and β actin (**d**). All data are given as mean ± S.E.M and were analysed via repeated measures one-way ANOVA with Sidak’s multiple comparisons test. (* *p* < 0.05; ** *p* < 0.01; *** *p* < 0.001; **** *p* < 0.0001 for inhibitor + FGF2 compared to FGF2-stimulated).

**Figure 6 ijms-25-06017-f006:**
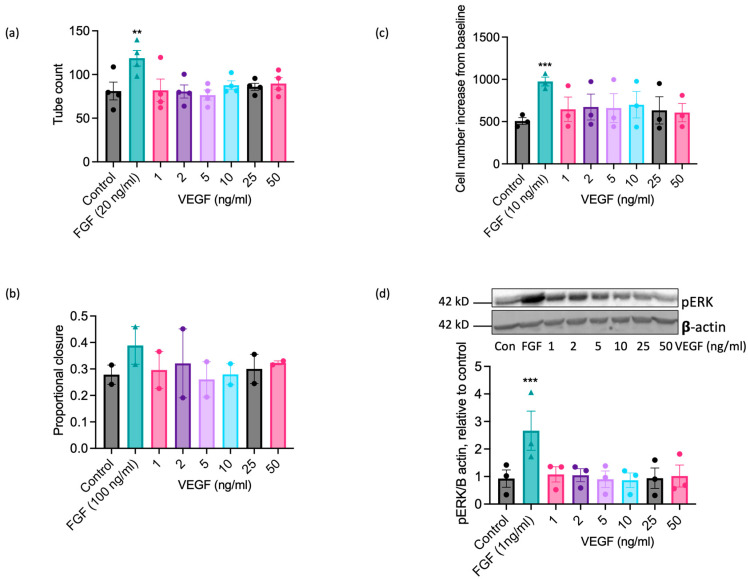
VEGF-A does not exert significant pro-angiogenic effects on EAoECs. EAoECs were exposed to FGF2 or VEGF-A at the indicated concentrations and (**a**) tube formation (16 h), (**b**) scratch wound closure (18 h), and (**c**) proliferation (72 h), assessed as described in the Materials and Methods Section 4. (**a**) Mean ± S.E.M., *n* = 4 (** *p* = 0.008 versus unstimulated control); (**b**) mean ± S.E.M. for *n* = 2; (**c**) mean ± S.E.M., *n* = 3 (*** *p* = 0.0003 versus unstimulated control). (**d**) ERK phosphorylation (10 min) was assessed via immunoblotting using a phospho-specific ERK1/2 antibody and β-actin as a loading control; blots were analysed via densitometry. Data are mean ± S.E.M. for *n* = 3 (*** *p* = 0.0006 versus unstimulated control). All statistical analysis used repeated measures one-way ANOVA with Dunnett’s multiple comparisons test.

**Figure 7 ijms-25-06017-f007:**
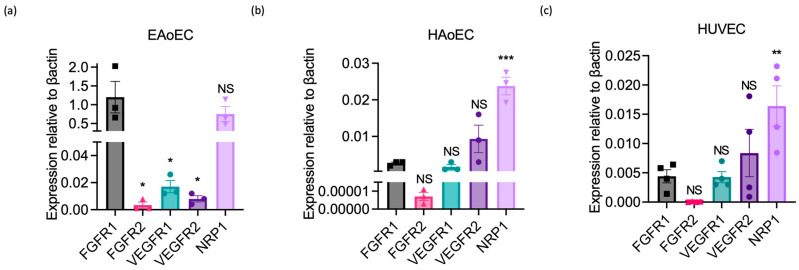
Growth factor receptor expression in equine versus human endothelial cells. Receptor expression was measured in unstimulated EAoECs, HAoECs, and HUVECs using qPCR and quantified relative to β-actin expression in the same cell type. (**a**) Relative FGFR1 expression in EAoECs is significantly greater than FGFR2 (* *p* = 0.01), VEGFR1 (* *p* = 0.01), and VEGFR2 (* *p* = 0.01). Mean ± S.E.M., *n* = 3). (**b**,**c**) Relative FGFR1 expression is significantly lower than NRP1 in HAoECs (*** *p* = 0.0002) and HUVECs (** *p* = 0.006). Data given are mean ± S.E.M of three and four biological repeats, respectively. All statistical analyses used repeated measures one-way ANOVA with Dunnett’s multiple comparisons test. (NS = not significant versus FGFR1, * *p* < 0.05, ** *p* < 0.01; *** *p* < 0.001).

## Data Availability

The data presented in this study are available on request from the corresponding author.

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
