# Peer review of "Equine Endothelial Cells Show Pro-Angiogenic Behaviours in Response to Fibroblast Growth Factor 2 but Not Vascular Endothelial Growth Factor A"

_ijms, 2024, doi:10.3390/ijms25116017_

Round 1

Reviewer 1 Report

Comments and Suggestions for Authors

The Manuscript by Finding and colleagues reports an optimised method of isolation and purification of equine aortic endothelial cells (EAoECs), and the effect of VEGF compared to FGF on growth and angiogenic function of these cells. Manuscript is very well written and the experiments performed and data provided are mostly solid. However, although the concept is interesting and potentially novel, the body of work collectively has not been sufficiently developed to significantly advance the field. In addition, there are some concerns in arriving at the conclusion that EAoECs do not respond to VEGF, which is the major conclusion of the study as discussed below.

Major concerns:

1)  The VEGF used for the studies was a recombinant equine growth factor, how confident are the authors that this recombinant growth factor is functional? Is there a possibility that a lack of effect EAoECs is due to impaired functionality of this recombinant molecule. Since human FGF was effective, it would be helpful to also test human VEGF A.

2) Also, in a related issue to the above point, the observation that VEGFR1/R2 mRNA levels are significantly lower than FGFR1, would not necessarily imply that cell surface expressions of VEGFR proteins are significantly reduced. As the author imply in the discussion that there may not be appropriate antibodies to evaluate these receptors at the protein levels, other approaches may be considered. For instance, could recombinant equine VEGFR2 be generated and used to overexpress in EAoECs and then transfectants treated with equine recombinant VEGF in the absence and/or presence of FGFR1 knockdown? Such an experiment may reveal that: 1) augmented expression of VEGFR does indeed increase response to VEGF, thus supporting the authors conclusion that it is the low level VEGFR receptor that is responsible for lack of response; 2) If there is response it will confirm that recombinant VEGFA is functional; 3) provides information regarding the effect of FGFR1 knock down on potentially causing compensatory switching the cell’s response to VEGF.

3) In Fig. 5, the use of MEK1/2 inhibitor PD184352 inhibited basal wound closure, proliferation and ERK1/2 phosphorylation, so it is difficult to conclude there was an effect on FGF-driven activities.

Minor concerns:

1)    In the introduction a brief rationale regarding why is there interest in using horse as a disease model would be helpful.

2)    In Figure 1, how accurately would IF staining determines VWF positivity in this assay? There may be variability in the levels of expression (high vs low) that could complicate arriving at conclusion that there is no VWF, while potentially there may be low VWF.

3)    The mechanical scraping also involves a collagenase treatment step as described in the legend describing Fig. 1d. How does this then ultimately affect the two processes?

4)    In figure 2e how were the %VWF positive cells were determined? Was it by FACS

5)    In Fig. 3a, although tubulogenesis by FGF appears to be statistically significantly higher than control and EGF; VEGF and IGF appear to be statistically not different from FGF. This requires further clarification.

Reviewer 2 Report

Comments and Suggestions for Authors

This is an elegant and well conducted study that contributes to to expand knowledge on endothelial pathophysiology.

I would like the authors to indicate the possible clinical repercussions in the management of atherosclerotic disease.

Comments on the Quality of English Language

minor revision

Reviewer 3 Report

Comments and Suggestions for Authors

In this manuscript, the authors described a optimized method to harvest and culture endothelial cells from equine aorta. They have also tested the angiogenic effects of these cells when treated with FGF-2 and VEGF-A.

Specific points:

1.      In a large number of the experiments shown, the "n" is 2, and there is no indication of repeated experiments. What was the power calculation that justified such a low number of replicates, and what assurance do the authors have of reproducibility?

2.      Figure 2 The quality of the fluorescent images in this figure are very low. 2a) the VE-cadherin staining is very punctate in the low magnification image, but in the high magnification image, the red staining is difficult to see.  2b) the image resolution is very low, difficult to see staining pattern.

3.      When describing “maximum” effect in a particular results, the author should reword the description. Unless the author can show significant differences between the different treatment groups, statistically, they are all the same.

4.      Fig 6, the author should consider adding a positive control in the experiments to make sure the VEGF-A is biologically active. This is to confirm that lack of pro-angiogenic effect on EAoECs were not due to technical problems.

5.      Methods: 4.4 Magnetic-activated cell sorting in supplementary methods should be added to this section. Since the one goal of this paper is to optimize a method to isolate and culture equine ECs.

6.      Supplementary: Table legends should be placed above the table.

Round 2

Reviewer 3 Report

Comments and Suggestions for Authors

Table legends should be above tables. The authors should review the "information for authors" and also look at published paper from IJMS for formatting. 

Otherwise there are no further comments.